# Relationship between Nutrition and Alcohol Consumption with Blood Pressure: The ESTEBAN Survey

**DOI:** 10.3390/nu11061433

**Published:** 2019-06-25

**Authors:** Alexandre Vallée, Amélie Gabet, Valérie Deschamps, Jacques Blacher, Valérie Olié

**Affiliations:** 1Paris-Descartes University, Diagnosis and Therapeutic Center, Hypertension and Cardiovascular Prevention Unit, Hôtel-Dieu Hospital, AP-HP, 75004 Paris, France; jacques.blacher@aphp.fr; 2Santé Publique France, The French Public Health Agency, 94410 Saint-Maurice, France; Amelie.GABET@santepubliquefrance.fr (A.G.); valerie.deschamps@univ-paris13.fr (V.D.); Valerie.OLIE@santepubliquefrance.fr (V.O.)

**Keywords:** DASH, blood pressure, diet, hypertension, alcohol

## Abstract

Background: Dietary interventions are recommended for the prevention of hypertension. The aim of this study was to evaluate and quantify the relationship between alcohol consumption and the DASH (Dietary Approaches to Stop Hypertension) score with blood pressure (BP) stratified by gender. Methods: Cross-sectional analyses were performed using data from 2105 adults from the ESTEBAN survey, a representative sample of the French population. Pearson correlation analyses were used to assess the correlation between the DASH score and alcohol with BP. Regressions were adjusted by age, treatment, socio-economic level, tobacco, exercise, Body mass index (BMI), and cardiovascular risk factors and diseases. Results: The DASH score was negatively correlated with systolic (SBP) and diastolic BP (DBP) (*p* < 0.0001). Alcohol was positively associated with increased BP only in men. The worst quintile of the DASH score was associated with an 1.8 mmHg increase in SBP and an 0.6 mmHg increase in SBP compared to the greatest quintile in men and with a 1.5 mmHg increase in SBP and an 0.4 mmHg increase in SBP in women. Male participants in the worst quintile of alcohol consumption showed an increase of 3.0 mmHg in SBP and 0.8 mmHg in DBP compared to those in the greatest quintile. Conclusion: A high DASH score and a reduction in alcohol consumption could be effective nutritional strategies for the prevention of hypertension.

## 1. Introduction

The prevalence of noncommunicable diseases has spread alarmingly around the world in both sexes and among all age subgroups [1]. Hypertension, one of the most important risk factors of noncommunicable diseases, is a major global public health problem affecting more than 30% of adults worldwide [2].

The major treatments prescribed for hypertension are dietary interventions, exercise, and medication [3]. Current guidelines for the prevention and treatment of hypertension emphasize lifestyle modifications which should enhance antihypertensive drug efficacy [3]. Dietary Approaches to Stop Hypertension (DASH) is a widely used dietary intervention for hypertension and is recommended by the American Heart Association. Previous studies have shown that the DASH diet lowers BP in those who normally have high blood pressure (BP) and has been recommended for the prevention and treatment of hypertension [4,5,6,7]. Moreover, when augmented by exercise and weight loss, the DASH diet can considerably benefit patients with high BP, not only by reducing BP but also by leading to favorable modification of disease risk biomarkers [8,9]. The DASH diet is rich in vegetables, fruits, low-fat dairy products, and legumes, and low in animal proteins. However, while many studies have investigated associations between diet and the incidence of hypertension, few studies have investigated the changes in BP and DASH diet indicators [4,5]. Indeed, several studies have shown a relationship between DASH and hypertension [6,10,11]. However, although there is strong and consistent evidence that adopting a DASH diet leads to a reduction in BP in hypertensive adults, the relationship between DASH and BP in the general population (hypertensive or not) is less reported [5,6,12,13]. Hypertension is a disease characterized by a persistent increase in BP. From this evidence, the control of arterial BP is important not only in hypertensive subjects but also in the general population [3]. Hypertension is largely a by-product of modern lifestyle determinants, including alcohol consumption. Recent guidelines recommend limiting daily alcohol intake to two or fewer drinks for men and 1 or fewer drinks for women [3]. Several studies have confirmed the association between heavy drinking and the development of hypertension [14]. However, this association remains unclear. The association may also depend on gender, which could be related to differential patterns [15].

Therefore, the aim of this study was to evaluate and quantify the relationships between alcohol consumption and the DASH score and its different components with parameters of blood pressure stratified by gender among a representative population of French adults.

## 2. Methods

### 2.1. Study Design

The ESTEBAN survey is a cross-sectional study that was conducted on a representative population of French adults. The protocol of the Esteban survey has been published previously [16]. One of the objectives of the Esteban survey was to describe food consumption, physical activity, sedentariness, and nutritional status and to estimate the prevalence of certain chronic diseases and vascular risk factors. The design of the Esteban survey was probabilistic at three degrees. In the first stage, a stratified sample of primary units was drawn. At the second level, in each primary unit, households were randomly selected by telephone sampling. At the third level, only one individual was randomly selected from the eligible household members. Stratification was carried out according to two variables: the region (8 geographical areas) and the degree of urbanization (5 strata: rural, <20,000 inhabitants, 20,000–100,000 inhabitants, >100,000 inhabitants, Paris). Data comprised dietary intake descriptions, clinical and biochemical marker measurements, physical activity, and complementary items assessed by questionnaires. The study was registered at the French National Agency for Medicines and Health Products Safety (No. 2012-A00456-34) and was approved by the Advisory Committee for Protection of Persons in Biomedical Research. 

### 2.2. Study Population

A total of 3021 adults was included between April 2014 and March 2016. After exclusion of participants, valid BP and adiposity-related measurements were available for 2105 participants who were included in the analyses (Figure 1). 

### 2.3. Dietary Intake Assessment

Dietary data were collected using three 24 h recalls, one of which was on the weekend, randomly distributed within a 2 week period. Participants were not previously informed of the call day to limit changes in eating habits for the survey. Specific questionnaires were used to investigate salt intake to describe the use of salt in cooking and salt added with a salt shaker during meals. Additional data were collected concerning the usual consumption of episodically consumed foods. In this paper, diet intake is reported based on the 24 h recalls (g/day). Consumption of alcoholic beverages was assessed using a combination of an annual propensity questionnaire and 24 h recalls (g/day). 

The DASH-style diet, as previously developed by Fung et al. [17], includes 8 dietary components which should be increased (fruits, vegetables, legumes, low-fat dairy, whole grains) or minimized (salt, sweetened beverages, red and processed meats). For each component, participants’ scores (ranging from 1 to 5) were based on sex-specific quintiles. The final DASH score, ranging from 8 to 40, was then obtained by adding the scores for each component. A high DASH score indicates a high intake of dietary fruits, vegetables, etc., and a low intake of salt, sweetened beverages, and meats. Quintiles of DASH scores were constructed to estimate the effect of the DASH classification on BP. Q1 was designated as the best quintile of DASH score and Q5 as the worst.

### 2.4. Blood Pressure and Other Anthropometric Measure Definitions

Body mass index (BMI) and blood pressure measurements were taken during the clinical examination based on standardized operational procedures. Weight and height were measured (respectively by a stadiometer fixed to a wall and Tanita scale with digital read-out), and BMI was calculated as body weight (kg) divided by the square of height (m).

Blood pressure was measured with an Omron 705-IT blood pressure monitor on the right arm using a cuff adapted to the circumference of the arm. Measurements were taken at 30 min from the blood test and after 5 min of rest, without change of position. Three measurements were made, 1 min apart. The systolic BP (SBP) and diastolic BP (DBP) for each person were taken the average of the last two measurements. People for whom at least two BP measurements were not completed were excluded from the analysis. Information on antihypertensive treatments were obtained by matching the individual data of the subjects included in the study with the data from the Sniiram (National inter-scheme health insurance information system). The names of the treatments and the reimbursement dates in the year preceding the health examination were collected.

### 2.5. Disease Definitions

Hypertension was defined as a systolic BP of ≥140 mm Hg and/or a mean diastolic BP of ≥90 mm Hg and/or antihypertensive treatment [3]. Hypercholesterolemia was considered if participants reported treating the condition or if the total cholesterol level was ≥6.61 mmol/L (255 mg/dL). Chronic kidney disease was defined as known proteinuria or decreased renal function (creatinine clearance <60 mL/min calculated by the Cockroft–Gault equation) for more than 3 months [18], or a chronic kidney disease diagnosed by biopsy or renal ultrasound and confirmed by a nephrologist. Participants were considered diabetic if they reported that they had been declared diabetic by a physician (or professional health worker) in the past or if they were currently using anti-diabetic treatment (oral agents or injections), or if they were declared diabetic by the physician in the health screening examination, or if their fasting blood glucose was ≥7 mmol/L. They were classified as non-diabetic otherwise. 

### 2.6. Socioeconomic Status 

Sociodemographic and economic data were collected by a questionnaire administered face-to-face during the first home visit. Duration of work was categorized in two classes: fixed-term contract and indefinite contract. Education level was determined according to the International Standard Classification of Education (ISCED) [19] and was then classified into three levels: High school diploma or less (≤13 years of education), undergraduate degree (14–16 years of education), and postgraduate degree (≥17 years of education). From the baseline questionnaires, we used the following sociodemographic variables: marital status (i.e., single (single/separated/widowed or divorced, and couple: marital or non-marital) and household income (i.e., <1600; 1600–2500; 2500–4600; >4600 euros per month). 

### 2.7. Statistical Analysis

For the participants who underwent a health examination, the probability of inclusion was calculated to consider the complex survey design. In addition, to consider a potential participation bias, calibration by age, education diploma, and the presence of at least one child in the household, according to French national census data and the period of collection, was performed separately for each gender. Calibration was carried out using the SAS macro program CALMAR (CALibration on MARgins), and census data came from the National Institute for Statistics and Economic Studies (Insee). Descriptive analyses were performed for the entire population and for each gender using counts and percentages or means ± standard deviations (SDs) for quantitative variables, or frequencies and percentages for categorical variables for the entire population and for each gender. For categorical variables, we used the Pearson’s Chi-square or Fisher’s exact tests, where applicable. Continuous quantitative variables were analyzed using the student (independent) *t*-test and Mann–Whitney test when a normal or abnormal distribution was assumed, respectively. BP parameters were SBP and DBP. DASH components were fruits, vegetables, low-fat dairy, milk, legumes, salt, sweetened beverages, and red and processed meats. Relationships were determined using linear regression models to assess the associations between covariables of interest and BP parameters. To investigate the determinants of BP, we used a multiple linear regression for each gender for SBP and DBP. Model-evaluated DASH scores and alcohol consumption were adjusted by age, education level, income level, term of contract, BMI, physical activity, hypercholesterolemia, diabetes, chronic kidney disease, previous cardiovascular (CV) disease, and antihypertensive therapy. First, we used a linear regression to evaluate the association between the DASH score and alcohol consumption with BP parameters (as continuous variables) and second, in same models, we included all the nutrient parameters of the DASH score and alcohol consumption to evaluate the independent determinants of BP parameters. Then, quintiles of the DASH score and alcohol consumption were constructed to evaluate their effects on BP gain or loss. The results were reported for each model as the square partial correlation coefficient, *r*^2^, which was used to describe the contribution of each parameter to BP variability. All tests were 2-sided; *p* < 0.05 was considered statistically significant. Statistical analyses were done using SAS software (version 9.4; SAS Institute, Carry, NC, USA).

## 3. Results

The characteristics of the 2105 included participants (945 men and 1160 women) stratified by gender are displayed in Table 1. The mean age of the study population (±SD) was 47.2 ± 14.6 years. The mean BMI (±SD) was 25.9 ± 5.1 kg/m^2^. The mean SBP (±SD) was 127 ± 19 mmHg. Smokers represented 23.9% of the participants. A total of 32.3% of our study population had a postgraduate degree. Among the 2105 participants, the prevalence of hypertension was 31.3%, 38.1% for men and 25.0% for women. Antihypertensive medication was taken by 48.9% of the hypertensive participants.

Hypertensive participants displayed higher BMI values (28.2 ± 5.7 kg/m^2^) than those were normotensive (24.9 ± 4.4 kg/m^2^, *p* < 0.0001), had higher daily alcohol consumption (10.2 ± 11.3 vs. 7.2 g ± 9.2, *p* < 0.0001), higher frequency of diabetes (12.4% vs. 3.7%, *p* = 0.004), higher frequency of hypercholesterolemia (32.8% vs. 15.9%, *p* < 0.0001) and lower DASH scores (23 ± 4 vs. 24 ± 4, *p* = 0.0002). 

In the multiple linear regression models, DASH score was independently and negatively associated with the two blood pressure parameters (SBP, DBP) in both genders (*p* < 0.0001) (Table 2). 

In men, multiple linear regression models of the decomposed components of the DASH score, for SBP: legumes (*p* = 0.01) and whole grains (*p* = 0.007) were independently and negatively associated with SBP whereas meats (*p* < 0.001) and sugary beverages (*p* = 0.003) were positively associated with SBP (Table 3). 

Likewise, for men, the consumption of legumes and wholegrains was negatively and independently associated with DBP, while the consumption of meats and sugary beverages was positively and independently associated with DBP.

In women, the multiple linear regression models of the decomposed components of the DASH score for SBP showed that legumes (*p* = 0.0001) were independently and negatively associated with SBP, whereas meats (*p* < 0.001), sugary beverages (*p* = 0.003), and salt (*p* = 0.0002) were positively associated with SBP. 

Likewise, for women, legumes were negatively and independently associated with DBP, while meats, sugary beverages, and salt were positively and independently associated with DBP.

Male participants with DASH scores in the worst quintile (Q5) had an average SBP of 128.8 mmHg and a DBP of 77.7 mmHg compared to male participants with DASH scores in the highest quintile (Q1), who had an average SBP of 127.0 mmHg and a DBP of 77.1 mmHg (in other words, an increase of 1.8 mmHg in SBP and 0.6 mmHg in DBP) (Figure 2, Table 4). Female participants with DASH scores in the worst quintile (Q5) had an average SBP of 126.9 mmHg and a DBP of 77.0 mmHg compared to male participants with DASH scores in the highest quintile (Q1), who had an average SBP of 125.4 mmHg and a DBP of 76.6 mmHg (in other words, an increase of 1.5 mmHg in SBP and 0.4 mmHg in DBP) (Figure 3, Table 4). 

In the multiple linear regression models, alcohol consumption was independently and positively associated with increased SBP and DBP values in men (respectively, *p* = 0.007 and *p* = 0.009) but not in women. Male participants with the lowest quintile of alcohol consumption had an average SBP of 128.6 mmHg and a DBP of 77.6 mmHg compared to male participants with the highest quintile of alcohol consumption, who had an average SBP of 125.6 mmHg and a DBP of 76.8 mmHg (in other words, an increase of 3.0 mmHg in SBP and 0.8 mmHg in DBP) (Figure 2, Table 4). 

## 4. Discussion

The main result of our study was that, in both genders, the DASH score was independently and negatively associated with BP variation, whereas alcohol consumption was only correlated with BP variation in men. Moreover, some differences appeared based on gender in the relationships between BP and the components of DASH score.

### 4.1. DASH Score and Blood Pressure

Two recent meta-analyses showed that the DASH diet significantly reduces BP. Saneei et al. showed that the DASH diet significantly reduces SBP by 6.74 mmHg and DBP by 3.54 mmHg [13], and Ndanuko et al. showed that the DASH diet significantly reduces SBP by 4.90 mmHg and DBP by 2.63 mmHg [20]. Some observational studies have reported a negative association between the DASH diet and blood pressure [12,21,22] which is consistent with our results. In a recent study among French volunteers (excluding hypertensive subjects undergoing vasoactive treatment), the DASH pattern was as follows: in females, the DASH diet was inversely associated with SBP, whereas for males, this association was not significant [6]. In contrast, in a cross-sectional study conducted in the framework of the Irish Mitchelstown Cohort Study, a significant inverse relationship emerged between adherence to the DASH diet and SBP in both genders [21]. Moreover, in the SU.VI.MAX cohort (individuals reporting antihypertensive treatment excluded), the DASH score was correlated with lower SBP and DBP at baseline and a lower BP increase after a follow-up period of 5 years [22]. In a European cohort (SUN (Seguimiento Universidad deNavarra) prospective cohort), it was observed that higher adherence to the DASH diet was correlated with a lower risk of developing hypertension (*p* = 0.02) [11]. On the contrary, a European cohort found no dose–response correlation between diet score and hypertension. This could be explained by the fact that the DASH scores in that study were derived from only three food components (vegetables, fruits, and milk products) [23]. 

### 4.2. Components of the DASH Score and BP 

#### Animal Products

Recent studies have reported that the intake of animal products (red and processed meats) is correlated with an increased risk of hypertension [5,24]. The noxious content of these foods results from the process of cooking, which produces advanced glycoxidation end-products (AGEs) and heterocyclic amines (HCAs) that increase oxidative stress and inflammation, counteracting the potential beneficial effect of protein intake from these foods. The longitudinal Chicago Western Electric Study observed that men who consumed 0.5 to 1.5 cups/day vs. less than 0.5 cups/day of vegetables presented a 2 mmHg lower rise in SBP over seven years (*p* < 0.05), whereas the consumption of beef, veal, lamb, and poultry was associated with a greater SBP/DBP increase (*p* < 0.05) [25]. Recently, an association between elevated fasting plasma trimethylamide N-oxide (TMAO) and an increased risk for main adverse CV events was identified [26]. TMAO is a metabolite of phosphatidylcholine and L-carnitine, which are both abundant in red meat. TMAO has been found to affect the hemodynamic effect of chronically infused angiotensin II [27], a pivotal hormone in circulatory system homeostasis. By affecting protein folding, TMAO affects the affinity of receptors as well as the activity of enzymes and hormones involved in the control of circulatory system homeostasis [26].

### 4.3. Gender Differences between Components of the DASH Score and BP 

#### 4.3.1. Salt Intake

Our study also identified a gender difference in the association between BP and nutritional factors. A few observational studies have reported this discrepancy but with inconsistent results [6,22,28]. We found that BP was associated with salt intake in women but not with men, when adjusted for all parameters. Several hypotheses can explain this finding. Data from a controlled dietary trial suggested that low dietary sodium intake may be more effective in reducing BP among women due to their higher salt sensitivity compared with men [29]. Dietary questionnaires only covered salt intake from foods. Total consumption was estimated by adding a constant for cooking meals and eating, which could have reduced the inter-individual variation and thus could made the potential correlation more difficult to identify. Another explanation is that the known relationship between BP and salt in the general population has been overstated and is more complex than once believed [28,30].

#### 4.3.2. Legumes and Wholegrains

The consumption of legumes is known to be correlated with lower BP levels [28,30,31]. This association could be explained by the high potassium and fiber contents of these vegetables, which have been found to reduce BP levels [32,33,34]. Our results are consistent with those in the literature in that we found a negative and independent correlation between BP parameters and the intake of legumes in both genders. 

The negative correlation between wholegrain consumption and BP can be explained by the ability of wholegrains to increase insulin sensitivity [31]. According to the results of our study, the association between dietary fiber and lower BP has been reported in several studies in men but not in women [5,35]. Moreover, a synergistic effect of fibers, i.e., wholegrains, with potassium and magnesium could improve mineral absorption in the gastrointestinal system [36,37]. 

There is significant evidence for a clear association between the microbiome and BP [38]. Indeed, hypertensive patients have different gut microbiome contents than individuals with normal BP [39]. Fecal transplant from hypertensive patients to gnotobiotic (germ free) mice led to an increase in BP (around 15 mmHg), gut metabolites resulting from the microbial fermentation of prebiotics are associated with lower BP values [39], and changes in the gut microbiome support the existence of a gut–cardio–renal axis and a gut–central nervous system axis [39,40]. The microbiome can be modulated by the intake of fiber, leading to a lower BP [40]. Moreover, this mechanism involves the production of short-chain fatty acids as by-products of the fermentation of fiber by intestinal bacteria [41].

#### 4.3.3. Sugar-Sweetened Beverages

Sugar-sweetened beverages (SSB) are the main source of added sugar in the diet. Many studies have presented a positive correlation between increased BP and consumption of SSB [42]. They reported, in general, that an intake of >12 fl of SSB per day can increase the risk of having hypertension by at least 6%, and it can increase MBP by a minimum of 1.8 mmHg in roughly 18 months [42]. However, it remains unclear at what dose increased SSB intake leads to the development of hypertension. The suggestion is that an intake of >1 serving of SSB per day is associated with a higher risk of hypertension [42]. High-fructose corn syrup (HFCS) is the main source of added sweeteners in sweetened beverages. A high intake of HFCS causes an increase in BP and thus involves an increase in uric acid production which, in turn, results in a lowering of nitric oxide in the body [43,44]. Other hypotheses behind high SSB intake being implicated in hypertension involve decreased sodium excretion, activation of the sympathetic nervous system [45,46], or an increased sodium concentration in the body through increased gut absorption [47]. Moreover, unhealthy lifestyle behaviors associated with increased sugar consumption by SSB go hand in hand with increased salt consumption leading to a higher BP [48,49].

#### 4.3.4. Alcohol Consumption

Alcohol consumption was associated with an increased BP in men. Notably, the strength of the association was distinct between both genders, and alcohol consumption did not reach significance in either gender and neither did alcohol consumption in women. In fact, high alcohol consumption and an increased BP and risk of hypertension have been described in several studies and in different populations [28,50]. Nevertheless, the relationship between hypertension and light-to-moderate alcohol consumption remains controversial in women [15]. A meta-analysis study assessed the presence of a gender-specific relationship between alcohol consumption and the risk of hypertension [14] and observed that high alcohol consumption was associated with the risk of hypertension in both genders, while moderate drinking was associated with a trend towards an increased risk of hypertension in men and a decreased risk in women. In our study, the difference in the magnitude of the effect among genders could be attributed to the differences in gender metabolism of drinking [51] and/or the patterns of drinking [52]. Accordingly, alcohol consumption should be limited in both genders [3]. 

Several studies have shown that ethanol consumption has a significant impact on blood pressure values, but this link remains complex [53]. Some mechanisms have been postulated for the hypertensive response to chronic ethanol consumption. Evidence suggests the existence of a myogenic mechanism involving alterations in the contractile and relaxant properties of vascular smooth muscle. Some studies have found that chronic ethanol consumption enhances the contractile response induced by phenylephrine in endothelium-intact aortic rings [54] and thus induces the contraction of blood vessels. Moreover, some studies have provided evidence that ethanol consumption alters plasma membrane permeability and then increases the intracellular Ca^2+^ concentration [53]. Importantly, ethanol consumption generates reactive oxygen species (ROS), which are a common mediator of remodeling and endothelial dysfunction [55]. Increased vascular oxidative stress induced by ethanol consumption is related to the activation of the enzyme nicotinamide adenine dinucleotide phosphatase (NAD(P)H) oxidase, and this pathway is associated with the increased blood pressure cause by chronic ethanol consumption [53]. Some evidence implicates the sympathetic nervous system, the renin–angiotensin–aldosterone system, increased intracellular Ca^2+^ in vascular smooth muscle, oxidative stress, decreased NO bioavailability, and endothelial dysfunction as underlying mechanisms that increase BP in association with ethanol consumption, but this issue remains an open one [53].

It should be noted that, as expected, in a previous French cohort, wine was the primary type of alcohol consumed (66% of alcohol consumption in both genders) [28]. However, in the Esteban survey, alcohol consumption was investigated in a global way without looking at the types of alcohol consumed. Alcohol consumption uniformly increases BP and CV risk events. Moderate alcohol intake could have a protective effect, but this link remains unclear [56]. 

#### 4.3.5. Strengths and Limitations

The main strength of our study is that participants of the ESTEBAN survey are representative of the general French population. The data were collected using standardized protocols which add validity to our study’s results. 

However, some limitations are present in our study. The self-administered questionnaire on weight history may have been subject to memory biases which would alter patient answers about social difficulties, recent financial limitations, refraining from care, weight variation, and age at which this variation occurred. Medical history and comorbidities were collected from insurance database notifications, self-reporting and physician assertions during medical examination in health centers. They were less subject to misinterpretation and under or over-reporting. The cross-sectional design of the study may represent a limitation. In cross-sectional analyses, reverse causation cannot be excluded. 

## 5. Conclusions

Approaches to the promotion of changes in lifestyle, including diet, are highly desirable to reduce BP levels and CV risk events or to avoid side effects related to chronic drug use. The DASH score, a general model of a health-conscious diet that is able to influence BP, could have implications in terms of public health and should be translated into dietary recommendations for the general population. Health professionals should continue to promote the consumption of vegetables and legumes due to the multiple effects of these food groups [57]. Red and processed meat consumption should be lowered. Even though no association has been observed between BP and alcohol consumption in women, health professionals should promote the lowering of excessive alcohol consumption level in both genders [58]. Moreover, recommendations based on dietary and drinking patterns may be a more comprehensive approach to the prevention of hypertension compared to single nutrition recommendations.

## Figures and Tables

**Figure 1 nutrients-11-01433-f001:**
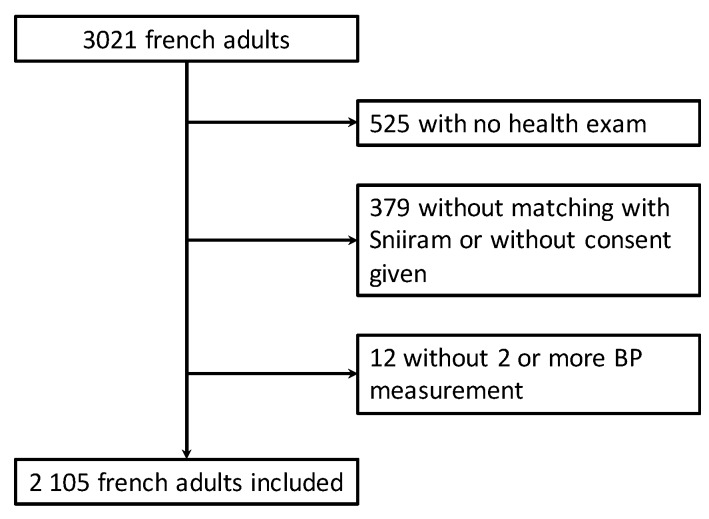
Flow chart, Esteban survey 2014–2016.

**Figure 2 nutrients-11-01433-f002:**
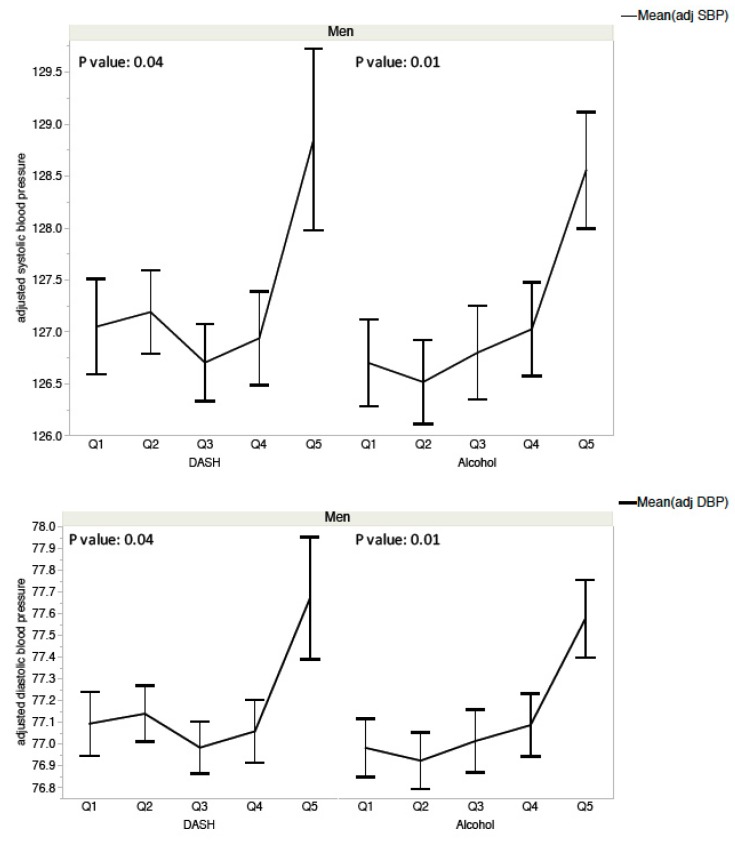
Blood pressure and DASH score quintiles and alcohol consumption in men from the Esteban survey 2014–2016. Q1: better quintile of DASH scores and alcohol consumption (lowest consumption).

**Figure 3 nutrients-11-01433-f003:**
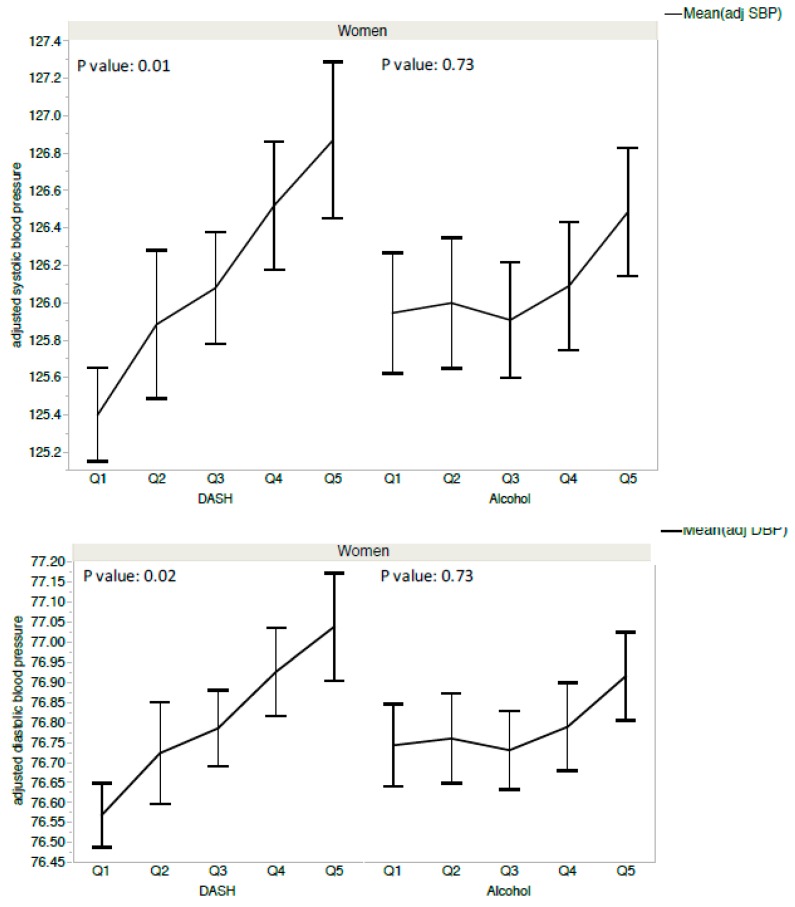
Blood pressure and quintiles of DASH score and alcohol consumption in women from the Esteban survey 2014–2016. Q1: better quintile of DASH scores and alcohol consumption (lowest consumption).

**Table 1 nutrients-11-01433-t001:** Characteristics of the study population, Esteban survey 2014–2016.

Characteristics	All	Men	Women	*p* Value *
*N*	2 105	945	1160	
Age (years)	47.2 (14.6)	47.7 (14.3)	46.7 (14.8)	0.61
Familial status				<0.0001
Couple	68.4%	72.7%	64.5%	
Single	31.6%	27.3%	35.5%	
Education level				<0.0001
<High school diploma	32.3%	32.0%	32.0%	
Undergraduate degree	56.5%	56.1%	56.9%	
Postgraduate degree	11.2%	11.3%	11.1%	
Income				<0.0001
Very high (>4200)	10.5%	12.6%	8.4%	
High	40.6%	39.9%	41.2%	
Medium	27.2%	28.2%	26.2%	
Low (<1600)	21.8%	19.3%	24.2%	
Contract				0.0003
Indefinite contract	84.9%	89.4%	80.9%	
Fixed-term contract	15.1%	10.6%	19.1%	
BMI (kg/m^2^)	25.9 (5.1)	26.1 (4.4)	25.7 (5.2)	<0.0001
Alcohol (g/day)	8.2 (9.9)	12.4 (11.9)	4.3 (5.3)	<0.0001
DASH score	23 (4)	23 (4)	24 (4)	<0.0001
Vegetables (g/day)	191.2 (71.6)	189.5 (73.2)	192.71 (70.2)	0.44
Legumes (g/day)	12.7 (6.1)	16.2 (6.6)	9.5 (3.7)	<0.0001
Red and processed meats (g/day)	126.1 (36.9)	148.5 (33.3)	105.4 (26.6)	<0.0001
Dairy products (g/day)	75.1 (106.9)	82.4 (120.3	68.3 (92.7)	0.0009
Wholegrains (g/day)	2.84 (3.87)	3.0 (3.9)	2.6 (3.8)	0.001
Juice and fruits (g/day)	220.4 (129.9)	224.7 (135.6)	216.5 (124.7)	0.09
Sweetened beverages (g/day)	76.3 (152.9)	87.6 (158.5)	65.9 (146.9)	<0.0001
Salt (g/day)	8.2 (2.7)	9.3 (2.8)	7.2 (2.2)	<0.0001
Current smoker	23.9%	27.1%	20.9%	<0.0001
Physical activity				
High	10.6%	15.6%	6.1%	<0.0001
Moderate	51.3%	56.2%	46.9%	
Low	38.1%	28.2%	47.0%	
Diabetes	6.5%	7.7%	5.4%	<0.0001
Hypercholesterolemia	21.2%	22.3%	20.1%	0.31
Chronic kidney disease	1.8%	1.3%	2.3%	0.04
Previous CV disease	4.5%	5.8%	3.2%	0.0002
Systolic BP	127 (19)	132 (18)	122 (18)	<0.0001
Diastolic BP	77 (11)	79 (11)	75 (10)	<0.0001
Pulse pressure	50 (13)	53 (13)	47 (12)	
Mean BP				
Hypertension	31.3%	38.1%	25.0%	<0.0001
Antihypertensive drugs **	48.9%	47.5%	51.1%	<0.0001

Values are means ± standard deviation, or numbers and % in parenthesis. BP: blood pressure, BMI: body mass index, CV: cardiovascular, DASH: Dietary Approaches to Stop Hypertension. * *p* value, difference between men and women. ** among hypertensives.

**Table 2 nutrients-11-01433-t002:** Multivariate linear regression of the DASH score and alcohol consumption for blood parameters.

			Men				Women		
		*r*^2^ value (%)	Estimate	Std	*p* value	*r*^2^ value (%)	Estimate	Std	*p* Value
SBP *	DASH score	0.14	−0.13	(0.07)	<0.0001	1.13	−0.61	(0.17)	<0.0001
DBP *		0.13	−0.10	(0.05)	<0.0001	1.12	−0.27	(0.10)	<0.0001
SBP **	Alcohol consumption	0.83	0.15	(0.03)	0.007	-	0.05	(0.05)	0.14
DBP **		0.77	0.11	(0.02)	0.009	-	0.01	(0.01)	0.16

* Model-evaluated DASH score with BP parameters were adjusted by age, education level, income level, term of contract, alcohol, tobacco use, physical activity, BMI, hypercholesterolemia, diabetes, chronic kidney disease, antihypertensive therapy, and previous CV disease. ** Model-evaluated alcohol consumption with BP parameters were adjusted by age, education level, income level, term of contract, DASH score, tobacco use, physical activity, BMI, hypercholesterolemia, diabetes, chronic kidney disease, antihypertensive therapy, and previous CV disease. DBP: diastolic blood pressure, SBP: systolic blood pressure.

**Table 3 nutrients-11-01433-t003:** Multivariate linear regression including the composed factors of DASH score and alcohol for blood parameters.

			Men				Women		
	**Parameters**	***r*^2^ value**	**Est.**	**Std**	***p* Value**	***r*^2^ value**	**Est.**	**Std**	***p* Value**
**Systolic blood pressure**	Vegetables	-	−0.01	(0.01)	0.44	-	−0.005	(0.01)	0.56
	Legumes	0.5	−0.31	(0.15)	0.001	1.8	−0.86	(0.22)	0.0001
	Red and processed meats	0.1	0.12	(0.03)	<0.0001	0.1	0.15	(0.03)	<0.0001
	Dairy products	-	−0.001	(0.008)	0.15	-	−0.005	(0.008)	0.35
	Wholegrains	0.4	−0.59	(0.21)	0.007	-	−0.39	(0.21)	0.06
	Juice and fruits	-	−0.003	(0.007)	0.58	-	−0.005	(0.008)	0.51
	Sweetened beverages	0.9	0.02	(0.008)	0.03	0.4	0.02	(0.007)	0.04
	Salt	-	0.03	(0.36)	0.92	0.1	1.63	(0.43)	0.0002
	Alcohol consumption	0.8	0.15	(0.2)	0.001	-	0.03	(0.06)	0.18
	**Parameters**	***r*^2^ value**	**Est.**	**Std**	***p* Value**	***r*^2^ value**	**Est.**	**Std**	***p* Value**
**Diastolic blood pressure**	Vegetables	-	−0.001	(0.009)	0.57	-	−0.0001	(0.008)	0.67
	Legumes	0.5	−0.19	(0.07)	0.008	1.9	−0.52	(0.13)	0.0001
	Red and processed meats	0.1	0.07	(0.01)	<0.0001	0.1	0.14	(0.01)	<0.0001
	Dairy products	-	−0.001	(0.005)	0.14	-	−0.09	(0.01)	0.34
	Wholegrains	0.4	−0.37	(0.12)	0.004	-	−0.25	(0.12)	0.05
	Juice and fruits	-	−0.001	(0.004)	0.72	-	−0.003	(0.005)	0.54
	Sweetened beverages	0.9	0.01	(0.004)	0.03	0.4	0.01	(0.004)	0.03
	Salt	-	0.68	(0.21)	0.75	0.1	0.99	(0.26)	0.0002
	Alcohol consumption	0.8	0.09	(0.05)	0.002	-	0.03	(0.08)	0.16

Model-evaluated nutrients and alcohol parameters were adjusted by age, education level, income level, term of contract, tobacco use, physical activity, BMI, hypercholesterolemia, diabetes, chronic kidney disease, antihypertensive therapy, and previous CV disease.

**Table 4 nutrients-11-01433-t004:** Adjusted BP according to quintiles of DASH scores and alcohol consumption in men and women. Q5: worst quintile of DASH scores and alcohol consumption (high) Q1: best quintile of DASH scores and alcohol consumption (lowest).

**DASH**	**MEN** **Score**	**Q5** **17 (2)**	**Q4** **21 (0.7)**	**Q3** **23 (.78)**	**Q2** **25 (0.7)**	**Q1** **28 (2)**	***p* value**
	SBP	128.8 (7.4)	126.9 (5.7)	126.7 (5.8)	127.1 (5.9)	127.0 (6.3)	0.04
	DBP	77.7 (2.4)	77.1 (1.8)	76.9 (1.8)	77.1 (1.9)	77.1 (2.0)	0.04
	**WOMEN** **score**	**Q5** **18 (2)**	**Q4** **22 (0.9)**	**Q3** **24 (0.8)**	**Q2** **27 (0.8)**	**Q1** **30 (2)**	***p* value**
	SBP	126.9 (5.6)	126.5 (5.3)	126.1 (5.0)	125.9 (4.9)	125.4 (3.9)	0.01
	DBP	77.0 (1.8)	76.9 (1.7)	76.8 (1.6)	76.7 (1.5)	76.6 (1.2)	0.02
**Alcohol**	**MEN** **g/day**	**Q5** **31.5 (8.0)**	**Q4** **17.2 (2.7)**	**Q3** **8.7 (1.8)**	**Q2** **3.9 (1.1)**	**Q1** **1.3 (0.9)**	***p* value**
	SBP	128.6 (7.3)	127.0 (5.9)	126.7 (5.9)	126.6 (5.3)	125.6 (5.5)	0.01
	DBP	77.6 (2.4)	77.1 (1.9)	77.1 (1.7)	76.9 (1.8)	76.8 (1.5)	0.01
	**WOMEN** **g/day**	**Q5** **14.3 (5.3)**	**Q4** **5.2 (1.2)**	**Q3** **2.5 (0.4)**	**Q2** **1.2 (0.2)**	**Q1** **0.5 (0.4)**	***p* value**
	SBP	126.5 (5.0)	126.1 (5.0)	125.9 (4.5)	126.0 (5.2)	125.9 (4.8)	0.74
	DBP	76.9 (1.5)	76.8 (1.6)	76.7 (1.5)	76.7 (1.6)	76.7 (1.6)	0.73

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
