# Peer review of "Relationship between Nutrition and Alcohol Consumption with Blood Pressure: The ESTEBAN Survey"

_nutrients, 2019, doi:10.3390/nu11061433_

Reviewer 1 Report

Summary: The DASH diet has been shown to lower blood pressure and has been recommended as a lifestyle intervention for the prevention and treatment of hypertension. The aim of this manuscript was to evaluate the relation between alcohol consumption and DASH scores using a cross sectional population of French adults (the Esteban survey). This is a well-designed study, and the authors have clearly stated the strengths and limitations.

Major criticisms

·         Methods: Most of the methods were well described; however, the dietary intake assessment section would benefit from more clarification.

1.     It would be helpful for the authors to state that a high DASH score indicates a high intake of dietary fruits, vegetables etc. and a low intake of salt, sweetened beverages and meats.

2.     It should also be stated in the second paragraph entitled “Dietary intake assessment” that the authors are designating Q1 as the best quintile of DASH scores (i.e., the highest DASH scores) and Q5 as the worst. It would be helpful to the reader if they further stated that their designation of quintiles is opposite to that used in the Fung paper, (cited by the authors, reference 17), in which Q1 was associated with the lowest DASH scores (i.e. a low intake of fruits and vegetables) and Q5 with the highest.

·         Results:

o    Table 3: Parts of the table do not line up correctly (dairy products, whole grains and juice and fruits in the column labeled Men).

o    Line 200: This is very confusing for the reader because the “lowest quintile of DASH scores” has been designated by the authors as Q5. It would be helpful to revise the sentence to say, “the worst quintile of DASH scores” . . .  (rather than “the lowest”) or, preferably “The lowest quintile of DASH scores (designated Q5)” .

Minor criticisms

·         Figure 1: The wording in the figure is unclear. Specifically, “379 patients with no appariement with Sniiram or no did not give their consent” and “12 with no 2 or more BP measurement.”

·         It would be helpful to get an estimate of alcohol consumption in the lowest and highest quintiles—in terms of g/day and equivalent in drinks/day. Could the lack of correlation between blood pressure and alcohol intake in women be explained by an overall intake of alcohol that is lower in women than in men? The authors recommend that “alcohol consumption should be limited by both genders”; however, their data do not support the recommendation that women should limit their intake of alcohol in order to lower their blood pressure.

Author Response

Answers to the Editor and Reviewers comments

The authors thank the Reviewers for their criticisms and suggestions. They have improved the form and substance of our manuscript. We hope to have answered all the questions asked. 

The additions and changes requested were written in red in the manuscript.

R1

Summary: The DASH diet has been shown to lower blood pressure and has been recommended as a lifestyle intervention for the prevention and treatment of hypertension. The aim of this manuscript was to evaluate the relation between alcohol consumption and DASH scores using a cross sectional population of French adults (the Esteban survey). This is a well-designed study, and the authors have clearly stated the strengths and limitations.

Major criticisms

·         Methods: Most of the methods were well described; however, the dietary intake assessment section would benefit from more clarification. 

1.     It would be helpful for the authors to state that a high DASH score indicates a high intake of dietary fruits, vegetables etc. and a low intake of salt, sweetened beverages and meats. 

The reviewer is right, we have added this information, in paragraph “Dietary intake assessment”.

2.     It should also be stated in the second paragraph entitled “Dietary intake assessment” that the authors are designating Q1 as the best quintile of DASH scores (i.e., the highest DASH scores) and Q5 as the worst. It would be helpful to the reader if they further stated that their designation of quintiles is opposite to that used in the Fung paper, (cited by the authors, reference 17), in which Q1 was associated with the lowest DASH scores (i.e. a low intake of fruits and vegetables) and Q5 with the highest.

According to the relevant observation of the reviewer, we have added this information in our manuscript, in dietary intake paragraph: “Quintiles of DASH score were constructed to estimate the effect of a classification of DASH on BP. Q1 designed the best quintile of DASH score and Q5 as the worst.”

·         Results:

·      Table 3: Parts of the table do not line up correctly (dairy products, whole grains and juice and fruits in the column labeled Men).

This has been corrected in table 3.

·      Line 200: This is very confusing for the reader because the “lowest quintile of DASH scores” has been designated by the authors as Q5. It would be helpful to revise the sentence to say, “the worst quintile of DASH scores” . . .  (rather than “the lowest”) or, preferably “The lowest quintile of DASH scores (designated Q5)” . 

According to the reviewer, we have change the term lowest by worst, line 200.  

Minor criticisms

·         Figure 1: The wording in the figure is unclear. Specifically, “379 patients with no appariement with Sniiram or no did not give their consent” and “12 with no 2 or more BP measurement.”

According to the reviewer, we have change the figure 1:

·         It would be helpful to get an estimate of alcohol consumption in the lowest and highest quintiles—in terms of g/day and equivalent in drinks/day. Could the lack of correlation between blood pressure and alcohol intake in women be explained by an overall intake of alcohol that is lower in women than in men? The authors recommend that “alcohol consumption should be limited by both genders”; however, their data do not support the recommendation that women should limit their intake of alcohol in order to lower their blood pressure.

According to the relevant observation of the reviewer we have added this information of g/day of alcohol consumption in Table 4. 

DASH

MEN

score 

Q5

17 (2)

Q4

21 (0.7)

Q3

23 (.78)

Q2

25 (0.7)

Q1

28 (2)

P value

SBP

128.8 (7.4)

126.9 (5.7)

126.7 (5.8)

127.1 (5.9)

127.0 (6.3)

0.04

DBP

77.7 (2.4)

77.1 (1.8)

76.9 (1.8)

77.1 (1.9)

77.1 (2.0)

0.04

WOMEN

score

Q5

18 (2)

Q4

22 (0.9)

Q3

24 (0.8)

Q2

27 (0.8)

Q1

30 (2)

P value

SBP

126.9 (5.6)

126.5 (5.3)

126.1 (5.0)

125.9 (4.9)

125.4 (3.9)

0.01

DBP

77.0 (1.8)

76.9 (1.7)

76.8 (1.6)

76.7 (1.5)

76.6 (1.2)

0.02

Alcohol

MEN

g/day

Q5

31.5 (8.0)

Q4

17.2 (2.7)

Q3

8.7 (1.8)

Q2

3.9 (1.1)

Q1

1.3 (0.9)

P value

SBP

128.6 (7.3)

127.0 (5.9)

126.7 (5.9)

126.6 (5.3)

125.6 (5.5)

0.01

DBP

77.6 (2.4)

77.1 (1.9)

77.1 (1.7)

76.9 (1.8)

76.8 (1.5)

0.01

WOMEN

g/day

Q5

14.3 (5.3)

Q4

5.2 (1.2)

Q3

2.5 (0.4)

Q2

1.2 (0.2)

Q1

0.5 (0.4)

P value

SBP

126.5 (5.0)

126.1 (5.0)

125.9 (4.5)

126.0 (5.2)

125.9 (4.8)

0.74

DBP

76.9 (1.5)

76.8 (1.6)

76.7 (1.5)

76.7 (1.6)

76.7 (1.6)

0.73

According to the relevant observation of the reviewer, we have change the conclusion as: 

“Approaches to promote changes in lifestyle, including diet, are highly desirable, aiming to reduce BP levels and then CV risk events or avoiding side effects related to chronic drug use. DASH score representing general models of health-conscious diets able to influence BP could have an implication in terms of public health and should be translated into dietary recommendations for the general population. Health professionals should continue to promote consumption of vegetables and legumes due to their multiple effects of these food groups [57], lowering red and processed meats. Even if no association has been observed between BP and alcohol in women, health professionals should promote the lowering of excessive alcohol consumption level in both genders [58]Moreover, recommendations based on dietary and drinking patterns may have a more comprehensive approach in the prevention of hypertension compared to single nutrition recommendation.”

Reviewer 2 Report

In this manuscript, entitled ”Relationship between Nutrition and Alcohol consumption with Blood Pressure: the ESTEBAN survey”, authors, Vallee et al, Analyzed DASH score and alcohol with BP in both men and women. They demonstrated that DASH score was negatively associated with BP and that alcohol was positively associated with increased BP in men. They concluded that High DASH score and reduced alcohol consumption could be effective nutritional strategies to prevent hypertension. The research area of this study (DASH-hypertension) is very interesting. However, there are several concerns in this study, which are listed in the following paragraphs:

1. DASH score on hypertension has been well-studied and reported by so many publications. What is the novel discovery in this research?

2. Dietary fibers could play a role in the effect of whole grains and vegetables on blood pressure. It would be better to provide this information in the Discussion section including the underlying mechanisms, such as gut microbiome, short chain fatty acids.

3. In this study, the relative novel part is the alcohol consumption on blood pressure.  What is the possible mechanism that mediates the effect of alcohol on the blood pressure?

4. Is the effect of alcohol consumption is related with doses of alcohol (how much alcohol consumed) or the type of alcohol? For example, there are reports showing that red wine may benefit for patients with hypertension and cardiovascular diseases. Please discuss this issue in the Discussion section

5. In the discussion, animal products, heterocyclic amine is mentioned (line 250). Heterocyclic amines should be HCAs instead of HAAs. In addition, more detail information regarding HCAs should be provided, such as TMAO and gut microbiomes.

Author Response

Answers to the Editor and Reviewers comments

The authors thank the Reviewers for their criticisms and suggestions. They have improved the form and substance of our manuscript. We hope to have answered all the questions asked. 

The additions and changes requested were written in red in the manuscript.

R2

In this manuscript, entitled ”Relationship between Nutrition and Alcohol consumption with Blood Pressure: the ESTEBAN survey”, authors, Vallee et al, Analyzed DASH score and alcohol with BP in both men and women. They demonstrated that DASH score was negatively associated with BP and that alcohol was positively associated with increased BP in men. They concluded that High DASH score and reduced alcohol consumption could be effective nutritional strategies to prevent hypertension. The research area of this study (DASH-hypertension) is very interesting. However, there are several concerns in this study, which are listed in the following paragraphs:

1.     DASH score on hypertension has been well-studied and reported by so many publications. What is the novel discovery in this research?

The reviewer is right, the association between DASH score and hypertension is well-known, however we think that the added value of our study was the representative sample of French population for this result.

2.     Dietary fibers could play a role in the effect of whole grains and vegetables on blood pressure. It would be better to provide this information in the Discussion section including the underlying mechanisms, such as gut microbiome, short chain fatty acids.

There is numerous evidence for a clear association between microbiome and BP [38]. Indeed, hypertensive patients have different gut microbiome compared to individuals with normal BP [39]; faecal transplants from hypertensive patients to gnotobiotic (germ free) mice lead to an increase in BP (around 15 mmHg); gut metabolites resultant of microbial fermentation of prebiotics are associated with lower BP [39]and changes in gut microbiome support the existence of a gut-cardio-renal axis and then a gut-central nervous system axis [39,40]. The microbiome can be modulated by intake of fibers leading in a lower BP [40]. Moreover, the mechanism involves the production of short-chain fatty acid as by-products of fermentation of fiber by intestinal bacteria [41].

3.     In this study, the relative novel part is the alcohol consumption on blood pressure.  What is the possible mechanism that mediates the effect of alcohol on the blood pressure?

Several studies have shown that ethanol-consumption has a significant impact on blood pressure values, but this link remains complex [53]. Somme mechanism have been postulated for the hypertensive response to chronic ethanol consumption. Evidence suggests the existence of a myogenic mechanism involving alterations in the contractile and relaxant properties of vascular smooth muscle. Some of studies have found that chronic ethanol consumption enhanced the contractile response induced by phenylephrine of endothelium-intact aortic rings [54]and thus induces contraction in blood vessels. Moreover, some studies have provided evidence that ethanol consumption alters the plasma membrane permeability and then increases the intracellular Ca2+ concentration [53]. Important ethanol consumption generates reactive oxygen species (ROS), which is a common mediator of remodeling and endothelial dysfunction [55]. Increased vascular oxidative stress induced by ethanol consumption is related to the activation of the enzyme nicotinamide adenine dinucleotide phosphatase (NAD(P)H) oxidase and this pathway is involved in the increased blood pressure cause by chronic ethanol consumption [53].Some evidences appear to implicate the sympathetic nervous system, the renin-angiotensin-aldosterone system, increased intracellular Ca2+in vascular smooth muscle, oxidative stress, decreased NO bioavailability and endothelial dysfunction as underlying mechanisms suggested to increase BP in association with ethanol consumption, but this issue remains an open one [53].

4.     Is the effect of alcohol consumption is related with doses of alcohol (how much alcohol consumed) or the type of alcohol? For example, there are reports showing that red wine may benefit for patients with hypertension and cardiovascular diseases. Please discuss this issue in the Discussion section

It should be noted that, as is expected in a previous French cohort, wine was the primary type of alcohol consumed (66% of alcohol consumption in both genders) [28]. However, in the Esteban survey, alcohol consumption was investigated in a global way without looking at the types of alcohol consumed. Alcohol consumption uniformly increases BP and CV risk events, even if moderate alcohol intake could presents apparently protective effects, but this link remains unclear [56]

5.     In the discussion, animal products, heterocyclic amine is mentioned (line 250). Heterocyclic amines should be HCAs instead of HAAs. In addition, more detail information regarding HCAs should be provided, such as TMAO and gut microbiomes.

HAAS has been change to HCAs

Recently, an association between an elevated fasting plasma trimethylamide N-oxide (TMAO) and an increased risk for main adverse CV events has been identified [26]. TMAO is a metabolite of phosphatidylcholine and L-carnitine, both abundant in red meat. TMAO has been found to affect the hemodynamic effect of chronically infused angiotensin II [27], a pivotal hormone in the circulatory system homeostasis. By affecting protein folding, TMAO affects the affinity of receptors as well as on the activity of enzymes and hormones involved in the control of the circulatory system homeostasis [26].

Round  2

Reviewer 2 Report

No other concern